# Copper(I)-Catalyzed Cross-Coupling of 1-Bromoalkynes with N-Heterocyclic Organozinc Reagents

**DOI:** 10.3390/molecules27144561

**Published:** 2022-07-17

**Authors:** Christian Frabitore, Jérome Lépeule, Tom Livinghouse

**Affiliations:** Department of Chemistry and Biochemistry, Montana State University, Bozeman, MT 59717, USA; christian.frabitore@student.montana.edu (C.F.); jerome.lepeule@student.montna.edu (J.L.)

**Keywords:** cross-coupling, copper, catalysis, heterocycles

## Abstract

Nitrogen-containing heterocycles represent the majority of FDA-approved small-molecule pharmaceuticals. Herein, we describe a synthetic method to produce saturated N-heterocyclic drug scaffolds with an internal alkyne for elaboration. The treatment of N,N-dimethylhydrazinoalkenes with Et_2_Zn, followed by a Cu(I)-catalyzed cross-coupling with 1-bromoalkynes, results in piperidines and pyrrolidines with a good yield. Five examples are reported and a proposed mechanism for the Cu(I)-catalyzed cross-coupling is presented.

## 1. Introduction

Nitrogen-containing heterocycles are recognized as privileged scaffolds for pharmaceuticals due to their increased pharmacokinetics and bioavailability [1,2,3]. As such, N-heterocycles are present in nearly 59% of all FDA-approved small molecule drugs and make up a plurality of newly approved drugs each year [4,5,6]. Piperidines and pyrrolidines appear in the top five most common N-heterocycles in FDA-approved drugs [1]. Industrial drug discovery initiatives often approach drug synthesis by creating large molecular libraries that can comprise hundreds of variations of one scaffold [7,8]. The use of neural networks and powerful algorithms to identify better and more specific target compounds has helped narrow the scope of these libraries [8]. However, each of these target compounds and their variants must be synthesized, purified, and assayed, which can take a significant amount of time to achieve for even a single drug candidate. Therefore, a synthetic method that provides stereodefined N-heterocyclic scaffolds with functional groups that can easily be modified would be extremely useful for drug discovery.

We have previously reported one such method involving the use of Et_2_Zn in a Zn(II)-mediated metalloamination/cyclization cascade with subsequent electrophilic functionalization (Figure 1) [9,10,11].

Unlike previously reported methods, the Zn(II)-mediated metalloamination/cyclization cascade does not rely on transient intermediates that limit the following functionalization to only those electrophiles which react quickly and completely [12,13,14,15,16,17,18,19,20]. The Zn(II) cascade generates a stable organozinc intermediate that can readily be transmetallated, resulting in the formation of a diverse repository of scaffolds. Here, we expand the scope of this method to the use of 1-bromoalkynes in Cu(I)-catalyzed functionalizations.

The cross-coupling of 1-haloalkynes with organozinc reagents was first reported by Knochel [21]. This initial report involved the coupling of alkyl organozincs with bromo- and iodoalkynes in the presence of catalytic CuCN·2LiCl to produce internal alkynes in good yields. The copper/zinc reagent RCu(CN)ZnX formed during this reaction is not well characterized, but subsequent studies by Knochel and others have elucidated the likely mechanism of this reaction (Figure 1, adapted from Thapa et al.) [22,23,24]. This original study demonstrated the usefulness of this method by its application to the synthesis of a pheromone present in *Amathes c-nigram* [21].

## 2. Results and Discussion

Our investigation started with N,N-dimethylhydrazinoalkenes **1–3** (Figure 2), which underwent metalloamination/cyclization as reported previously [9,11]. Benzotriflouride and toluene provided excellent conversion to the organozinc intermediate, with toluene proceeding at slower rate. In contrast, 1,2-dichlorethane, diisopropyl ether, acetonitrile, and isobutyronitrile were found to provide incomplete conversion.

These intermediates were treated with 1-bromo-1-octyne (two equiv.), CuCN·2LiBr (5 mol%), and LiBr (one equiv.). The utilization of LiBr as an additive confers a significant stabilizing and accelerating influence, as we have previously reported [9]. Ethereal solvents were necessary, with THF being optimal for Cu(I)-catalyzed electrophilic functionalization. The isolation of the products was straightforward and produced **1a** and **2a** in good yields (Table 1). Product **3a** demonstrates that an increased steric bulk on the N-heterocycle still leads to successful coupling. Having found success with these reactions, we moved forward to coupling with the more reactive alkyne, 1-bromo-2-phenylethyne.

Knochel reported that Cu(I)-catalyzed cross-couplings with 1-bromo-2-phenylethyne were difficult to control. Successful reactions with 1-bromo-2-phenylethyne took 1–4 h at −76 °C to complete, and increasing the temperature inevitably led to decreased yields [21]. Diamine ligands can exert a stabilizing effect on copper catalysts [25]. These ligands also accelerate Cu(I)-catalyzed cross-coupling reactions [26,27]. Unfortunately, in our hands, the coupling of **2** with 1-bromo-2-phenylethyne in the presence of TMEDA (one equiv.) led to a marked decrease in the yield of **2b** (25%). Lowering the reaction temperature led to no improvement. In this case, the product mixture contained a homocoupled diyne product and the Zn metalloamination intermediate derived from **2**. In consonance with our previously reported conditions [9], the inclusion of one equiv. of LiBr allowed the alkynyalation reactions leading to both **2a** and **2b** to be run at room temperature with improved yields.

## 3. Materials and Methods

### 3.1. Materials

All materials were either prepared according to previous literature or purchased from commercial sources. N,N-dimethylhydrazinoalkene substrates **1–3** were synthesized by previously reported methods [11,27] and stored in an inert atmosphere glovebox. All solvents were obtained from a JC Meyer solvent dispensing system. Inorganic salts were dried by heating under a high vacuum before use. Thin-layer chromatography (TLC) employed 0.25 mm glass silica gel plates with UV indicator and were visualized with UV light (254 nm) or potassium permanganate staining. Nuclear magnetic resonance (NMR) data were obtained from a Bruker AVANCE III HD NMR spectrometer equipped with an Ascend 500 (500 MHz) magnet. High-resolution mass spectra (HRMS) were obtained from a Bruker MicroTOF with a Dart 100–SVP 100 ion source.

### 3.2. Experimental Procedures

#### 3.2.1. General Procedure for the Synthesis of Alkynyl N,N-Dimethylhydrazinoalkenes

In an inert atmosphere glovebox, a 5 mm J. Young NMR tube with a sealed 3 mm NMR calibration tube containing C_6_D_6_ was charged with benzotrifluoride (BTF, 0.4 mL), 2 M diethyl zinc in BTF (60 μL, 0.12 mmol, 1.2 equiv.), and the requisite N,N-dimethylhydrazinoalkene (0.1 mmol). The J. Young tube was placed in a 90 °C oil bath and the cyclization reaction was monitored by NMR (indicated by the recession of the alkene peaks), using BTF as an internal standard. Upon completion of the cyclization, the volatiles were removed in vacuo. The J. Young NMR tube was returned to the glovebox and was charged with THF (0.3 mL), the requisite 1-bromoalkyne (0.2 mmol), 0.5 M CuCN·2LiBr solution in THF (10 μL, 5 mol%), and 2 M LiBr in THF (50 μL, 0.1 mmol, 1 equiv.). The reactant mixture was allowed to stand at room temperature overnight. The resulting solution was dispersed in diethyl ether (1 mL) and washed with an aqueous solution of 1:1 sat. NH_4_OH/sat. NH_4_SO_4_ (3 × 1 mL). The combined aqueous washes were then back-extracted with diethyl ether (2 × 1 mL) and the combined organic layers were concentrated in vacuo. The resulting crude product was dissolved in CH_2_Cl_2_ (1 mL), dispersed onto silica gel (50 mg), and dried in vacuo. This was purified by dry-loading the resulting silica gel onto a silica gel plug and eluting with pentane. The product was then flushed from the silica gel using diethyl ether and concentrated in vacuo. The product could be further purified, if necessary, by column chromatography (15% ether/pentane for elution). 

1-(N,N-dimethylamino)-2-(2-heptynyl)-5-methyl-2a-3,4-5a-tetrahydropyrrole (**1a**). The title compound was synthesized by the general procedure, producing 19.2 mg (77%) of the title compound as a clear to yellow oil. ^**1**^**H NMR** (CDCl_3_, 500 MHz): δ 3.04 (dquint, 1H, *J* = 8.2, 6.2 Hz), 3.02–2.95 (m, 1H), 2.55–2.49 (m, 1H), 2.48 (s, 6H), 2.19–2.13 (m, 2H), 2.10 (ddt, 1H, *J* = 16.2, 9.3, 2.4 Hz), 1.83 (dq, 1H, *J* = 12.4, 8.0 Hz), 1.78–1.70 (m, 1H), 1.64–1.55 (m, 1H), 1.53–1.45 (m, 2H), 1.44–1.25 (m, 7H), 1.13 (d, 3H, *J* = 6.2 Hz), 0.91 (t, 3H, *J* = 7.0 Hz). **^13^C NMR** (CDCl_3_, 126 MHz): δ 80.87, 78.79, 59.34, 54.56, 41.37, 31.53, 30.48, 29.28, 28.67, 27.84, 26.41, 22.73, 22.15, 18.95, 14.21. **HRMS** electrospray ionization (ESI) *m*/*z* calcd. for C_16_H_30_N_2_: 250.2409, found: 250.2415 (mass error Δm = 2.46 ppm). **^1^H NMR** and **^13^C NMR** spectra of product **1a** can be found in the Appendix A.

1-(N,N-dimethylamino)-2-(2-heptynyl)-2a-,3,4,5,6-pentahydropyridine (**2a**). The title compound was synthesized by the general procedure, producing 20.3 mg (81%) of the title compound as a clear to yellow oil. ^**1**^**H NMR** (CDCl_3_, 500 MHz): δ 2.99–2.94 (m, 1H), 2.91 (ddt, 1H, *J* = 16.4, 3.3, 2.5 Hz), 2.37–2.31 (ddt, 1H, *J* = 10.6, 9.0, 3.1 Hz), 2.29 (s, 6H), 2.17–2.10 (m, 3H), 2.05 (ddt, 1H, *J* = 16.4, 9.0, 2.5 Hz), 2.05–1.99 (m, 1H), 1.73–1.64 (m, 2H), 1.52–1.42 (m, 3H), 1.41–1.34 (m, 2H), 1.33–1.22 (m, 5H), 1.14 (qt, 1H, *J* = 13.2, 4.0 Hz), 0.88 (t, 3H, *J* = 7.0 Hz). **^13^C NMR** (CDCl_3_, 126 MHz): δ 81.50, 78.92, 59.82, 42.90, 39.14, 32.32, 31.53, 29.35, 28.71, 26.34, 24.74, 23.79, 22.73, 19.06, 14.19. **HRMS** electrospray ionization (ESI) *m*/*z* calcd. for C_16_H_30_N_2_: 250.2409, found: 250.2412 (mass error Δm = 1.00 ppm). **^1^H NMR** and **^13^C NMR** spectra of product **2a** can be found in the Appendix A.

1-(N,N-dimethylamino)-2-(2-heptynyl)-4,4-dimethyl-2a-3,5-tetrahydropyrrole (**3a**). The title compound was synthesized by the general procedure, producing 15.3 mg (58%) of the title compound as a clear to yellow oil. ^**1**^**H NMR** (CDCl_3_, 500 MHz): δ 2.85 (ddd, 1H, *J* = 8.9, 8.0, 3.3 Hz) 2.6 (d, 1H, *J* = 8.2 Hz), 2.62 (ddt, 1H, J = 16.2, 3.3, 2.4 Hz), 2.44 (d, 1H, *J* = 8.2 Hz), 2.35 (s, 6H), 2.15 (m, 2H), 2.09 (ddt, 1H, *J* = 16.2, 8.9, 2.4 Hz), 1.69 (dd, 1H, *J* = 8.0, 12.6 Hz), 1.52–1.45 (m, 2H), 1.44–1.36 (m, 3H), 1.36–1.25 (m, 4H), 1.12 (s, 3H), 1.05 (s, 3H), 0.91 (t, 3H, *J* = 7.1 Hz). **^13^C NMR** (CDCl_3_, 126 MHz): δ 80.88, 78.73, 59.92, 54.30, 43.97, 40.46, 34.06, 31.54, 30.54, 29.40, 29.30, 28.67, 24.44, 22.75, 18.97, 14.21. **HRMS** electrospray ionization (ESI) *m*/*z* calcd. for C_17_H_32_N_2_: 264.2565, found: 264.2567 (mass error Δm = 0.69 ppm). **^1^H NMR** and **^13^C NMR** spectra of product **3a** can be found in the Appendix A.

1-(N,N-dimethylamino)-2-(3-phenyl-2-propynyl)-5-methyl-2a-3,4-5a-tetrahydropyrrole (**1b**). The title compound was synthesized by the general procedure, producing 16.0 mg (66%) of the title compound as a clear to yellow oil. ^**1**^**H NMR** (CDCl_3_, 500 MHz): δ 7.44–7.40 (m, 2H), 7.32–7.25 (m, 3H), 3.15–3.06 (m, 2H), 2.77 (dd, 1H, *J* = 16.6, 3.7 Hz), 2.53 (s, 6H), 2.39 (dd, 1H, *J* = 16.6, 9.0 Hz), 1.95–1.85 (m, 1H), 1.83–1.74 (m, 1H), 1.74–1.65 (m, 1H), 1.47–1.37 (m, 1H), (d, 3H, *J* = 6.0 Hz). **^13^C NMR** (CDCl_3_, 126 MHz): δ 131.70, 128.29, 127.54, 124.36, 89.23, 81.15, 59.28, 54.29, 41.43, 30.60, 27.86, 26.93, 22.27. **HRMS** electrospray ionization (ESI) *m*/*z* calcd. for C_16_H_22_N_2_: 242.1783, found: 242.1784 (mass error Δm = 0.48 ppm). **^1^H NMR** and **^13^C NMR** spectra of product **1b** can be found in the Appendix A.

1-(N,N-dimethylamino)-2-(3-phenyl-2-propynyl)-2a-3,4,5,6-pentahydropyridine (**2b**). The title compound was synthesized by the general procedure, producing 17.4 mg (72%) of the title compound as a clear to yellow oil. ^**1**^**H NMR** (CDCl_3_, 500 MHz): δ 7.40–7.45 (m, 2H), 7.24–7.32 (m, 3H), 3.18 (dd, 1H, *J* = 16.7, 3.5 Hz), 2.99–3.05 (m, 1H), 2.48–2.55 (m, 1H), 2.36 (s, 6H), 2.36 (dd, 1H, *J* = 16.6, 8.9 Hz), 2.20 (ddd, 1H, *J* = 12.3, 10.6, 2.5 Hz), 2.08–2.15 (m, 1H), 1.69–1.78 (m, 2H), 1.47–1.58 (m, 1H), 1.34–1.44 (m, 1H), 1.15–1.26 (m, 1H). **^13^C NMR** (CDCl_3_, 126 MHz): δ 131.71, 128.27, 127.44, 124.55, 89.66, 81.78, 59.65, 42.93, 39.27, 32.52, 26.33, 24.75, 24.60. **HRMS** electrospray ionization (ESI) *m*/*z* calcd. for C_16_H_22_N_2_: 242.1783, found: 242.1784 (mass error Δm = 0.62 ppm). **^1^H NMR** and **^13^C NMR** spectra of product **2b** can be found in the Appendix A.

#### 3.2.2. Procedure for Preparative Scale Synthesis of **3a**

1-(N,N-dimethylamino)-2-(2-heptynyl)-4,4-dimethyl-2a-3,5-tetrahydropyrrole (**3a**). In an inert atmosphere glovebox, a 5 mm J. Young NMR tube was charged with 0.4 mL of benzotrifluoride, 300 μL of 2-(2,2-dimethylpent-4-en-1-yl)-1,1-dimethylhydrazine (1.5 mmol), and 185 μL of Et2Zn (1.8 mmol, 1.2 equiv.). The J. Young tube was placed in a 90 °C oil bath and the cyclization reaction was monitored by No-D NMR (indicated by the recession of the alkene peaks) using BTF as an internal standard. Upon completion of the cyclization, the J. Young tube was returned to the glovebox and the contents were transferred to a 10 mL Schlenk flask equipped with a magnetic stirring bar, and a Schlenk adapter with a septum on the sidearm. The J. Young tube was rinsed with THF (3 × 0.5 mL), which was added to the Schlenk flask, and the flask was then removed from the glovebox. The volatiles were removed in vacuo and the resulting solids were redissolved in a preformed solution of 500 μL of 1-bromo-1-octyne (3.125 mmol, 2.08 equiv.) in 1.5 mL of THF. The resulting solution was cooled to −78 °C and a preformed solution of 130 mg of LiBr (1.5 mmol, 1 equiv.) and 150 μL of 0.5 M CuCN·2LiBr solution in THF (5 mol%) in 3 mL of THF was slowly added. The solution was allowed to warm to RT with stirring overnight. The reactant mixture was then dispersed in 5 mL of diethyl ether and washed with a 1:1 sat. aqueous NH_4_Cl: sat. aqueous NH_4_OH solution (3 × 2 mL). The combined aqueous layers were back-extracted with diethyl ether (2 × 2 mL). The combined organic layers were dried with brine and Na_2_SO_4_, concentrated in vacuo, and the volatile components were removed under a high vacuum. The crude product was further purified by column chromatography using gradient elution (hexane to 20% EtOAc/hexane) to provide 293.1 mg (76%) of the title compound as a yellow oil. **^1^H NMR** (CDCl_3_, 500 MHz): δ 2.85 (ddd, 1H, *J* = 8.9, 8.0, 3.3 Hz) 2.6 (d, 1H, *J* = 8.2 Hz), 2.62 (ddt, 1H, J = 16.2, 3.3, 2.4 Hz), 2.44 (d, 1H, *J* = 8.2 Hz), 2.35 (s, 6H), 2.15 (m, 2H), 2.09 (ddt, 1H, *J* = 16.2, 8.9, 2.4 Hz), 1.69 (dd, 1H, *J* = 8.0, 12.6 Hz), 1.52–1.45 (m, 2H), 1.44–1.36 (m, 3H), 1.36–1.25 (m, 4H), 1.12 (s, 3H), 1.05 (s, 3H), 0.91 (t, 3H, *J* = 7.1 Hz). **^13^C NMR** (CDCl_3_, 126 MHz): δ 80.88, 78.73, 59.92, 54.30, 43.97, 40.46, 34.06, 31.54, 30.54, 29.40, 29.30, 28.67, 24.44, 22.75, 18.97, 14.21. **HRMS** electrospray ionization (ESI) *m*/*z* calcd. for C_17_H_32_N_2_: 264.2565, found: 264.2567 (mass error Δm = 0.69 ppm). **^1^H NMR** and **^13^C NMR** spectra of product **3a** can be found in the Appendix A.

#### 3.2.3. General Procedure for the Synthesis of 1-Bromoalkynes

To a 100 mL round bottomed flask equipped with a magnetic stirring bar, acetone (30 mL) and the requisite 1-alkyne (12.5 mmol) were added. To this solution, AgNO_3_ (0.213 g, 1.25 mmol) was added and, with vigorous stirring, N-bromosuccinimide (2.5 g, 14 mmol) was added in portions. The mixture was stirred at room temperature for 2 h when confirmed to be completed by TLC. The reactant mixture was diluted in pentane (75 mL) and filtered. The resulting solution was washed with water (2 × 25 mL) and the combined aqueous washes were back-extracted with 1:1 diethyl ether/pentane (2 × 15 mL). The combined organic layers were dried with Na_2_SO_4_, filtered through a plug of silica gel, and concentrated in vacuo. The product could be further purified by vacuum distillation or flash column chromatography. Procedure adapted from Gao et al. [28].

1-Bromo-1-octyne. The title compound was synthesized by the general procedure. Short-path distillation in vacuo over CaH_2_ produced 2.10 g (89%) of the title compound as a clear liquid. **^1^H NMR** (C_6_D_6_, 500 MHz): δ, 1.87 (t, 2H, *J* = 6.9 Hz), 1.19–1.27 (m, 2H), 1.10–1.19 (m, 4H), 1.00–1.08 (m, 2H), 0.80 (t, 3H, *J* = 7.2 Hz). **^13^C NMR** (C_6_D_6_, 126 MHz): δ 80.81. 38.26, 31.53, 28.69, 28.50, 22.82, 19.80, 14.21. **HRMS** data has been previously reported [29]. **^1^H NMR** and **^13^C NMR** spectra of 1-bromo-1-octyne can be found in the Appendix A.

1-Bromo-2-phenylethyne. The title compound was synthesized by the general procedure. Flash column chromatography with 1:1 ether/pentane produced 2.24 g (99%) of the title compound as a clear liquid. **^1^H NMR** (C_6_D_6_, 500 MHz): δ 7.24–7.27 (m, 2H), 6.83–6.92 (m, 3H). **^13^C NMR** (C_6_D_6_, 126 MHz): δ 132.28, 128.80, 128.55, 123.13, 80.73, 50.49. **HRMS** data has been previously reported [29]. **^1^H NMR** and **^13^C NMR** spectra of 1-bromo-2-phenylethyne can be found in the Appendix A.

## 4. Conclusions

The Cu(I)-catalyzed cross-coupling of 1-bromoalkynes with the aminozincation intermediates derived from N,N-dimethylhydrazinoalkenes is described here. The use of LiBr as an adjuvant led to higher conversions and improved yields in this transformation. The incorporation of TMEDA (1.0 equiv.) was detrimental in the cross-couplings involving 1-bromo-2-phenylethyne. It is significant from a practical standpoint that this method can easily be extended to a preparative scale with no decrease in yield. The present technique provides saturated N-heterocyclic pharmaceutical scaffolds possessing a highly modifiable internal alkyne for further synthetic elaboration. The extension of this method to the utilization of heteroatom-bearing 1-bromoalkynes, as well as alternative N,N-dimethylhydrazinoalkenes, will be the topic of future reports from this laboratory.

## Data Availability

Not applicable.

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
