# Peer review of "Copper(I)-Catalyzed Cross-Coupling of 1-Bromoalkynes with N-Heterocyclic Organozinc Reagents"

_molecules, 2022, doi:10.3390/molecules27144561_

Round 1

Reviewer 1 Report

The authors report the synthesis of piperidines and pyrrolidines via sequential treatment of N,N-dimethylhydrazinoalkenes with Et2Zn and a Cu(I) catalyzed cross-coupling with 1-bromoalkynes. This is a well-written and executed study worthy of publication. I have a few suggestions/corrections for the authors to address:

1)      It is better to provide the reaction conditions as a footnote under the Table 1, also indicating the results under different conditions in Table 1 (for example, the lowering temperature… what temperature? what yields? etc). Giving, just the information in the sentence (for example, lines 78, 80) is not informative for the readers.

2)      Did the authors perform the reaction in a normal flask under Schlenk line and in a bigger scale (for example, 1 or 10 mmol scales)? It is interesting to see the reproducibility on scale up reaction and that, in future, the reaction can be run in any lab, and not just in glove box and on 0.1 mmol scale.

3)      For the products 1a and 1b it is better to indicate that they formed as cis-isomers in racemic forms, otherwise the drawn structures show that the obtained products seems to be enantiopure compounds.

4)      In the SI file, NMR spectra better to provide as landscape format to facilitate their interpretation.

5)      Ref. 6, the second author’s name should be as initial.

6)      Ref. 16, no journal name.

7)      Ref. 20, line 245, delete one “P” in the word “PPyrrolidinones”.

Author Response

First, we thank the Reviewer for their time and attention given to important details.

Comment 1: The figure above Table 1 has been modified to reflect the temperature and time of reaction. Footnotes have been added to delineate the exact reaction conditions.

Comment 2: A preparative scale reaction leading to product 3a has been reported alongside the 0.1 mmol scale reaction. A procedure for this reaction has been added.

Comment 3: A footnote has been provided for Table 1 to indicate that the products are mixtures of both cis isomers.

Comments 4-7: The designated errors were corrected.

Reviewer 2 Report

The submitted article ‘Copper(I) Catalyzed Cross-Coupling of 1-Bromoalkynes with N-Heterocyclic Organozinc Reagents’ is a very short communication. The topic is interesting, but written in a very concise way. The results and discussion part states only 1 page out of 7. Chapter 3 is missing, followed by chapter 4 - Materials and methods, described in detail. I also do not find any conclusions. Information on good yield is included in the abstract. Is the yield of 3a and 2b satisfactory? To sum up, although it is a communication, the part of the manuscript related to results and discussions should be more detailed and conclusions should be added.

Author Response

First, we thank the Reviewer for their time and attention given to important details.

We have performed selected reactions under modified conditions to obtain improved yields and have added a concise discussion of this. The error in the missing chapter was corrected and a Conclusion section was added.

Reviewer 3 Report

This paper deals with the synthesis of N-heterocycles such as piperidines and pyrrolidines with an internal alkyne via Cu(I) catalyzed cross-coupling and metal mediated cyclization protocols. In my opinion described studies could be an interesting extension of the current knowledge. Although, most of the work expressed in the submitted MS have already been described by the authors and other groups in previous papers. In my opinion, overall amount of novelty is not remarkably high, however, ample to allow publication, after addressing a
few concerns.

Remarks to authors

1.     Authors should describe similar study in detail, earlier and present report in Scheme 1 with proper citations.

2.     Some more examples need to be included. Did authors try with hetero alkynes and dimethyl derivative such as 3??

3.     In page 2/3, separate scheme should be provided and table 1 should be improve with better manner with all conditions and scheme. R/R, = ?? n = ??, time vs temp??

4.     I strongly suggest authors to include some of them DEPT for complete characterizations.

5.     In line 84/page 3—section 4 mentioned but I did not see section 3 and conclusions of MS.

6.      I can see that some of the compounds NMR data not consistent with spectrums, total expected, please double check carefully for <ex> in SI file compound 1b/13C value 54.14 is it compound peak?

Author Response

First, we thank the Reviewer for their time and attention given to important details.

Comment 1: ‘Authors should describe similar studies in detail earlier and present report in Scheme 1 with proper citations.’ This has been done (see manuscript, lines 59-60).

Comment 2: We have included a preparative scale procedure and yield for product 3a. Heteroalkynes and additional N,N-dimethylhydrazinoalkenes are currently being explored. These results are preliminary and will be reported in an upcoming full paper.

Comment 3: Table 1 has been modified by the inclusion of a new figure and footnotes describing the reaction conditions.

Comment 5: The section numbering error has been corrected and a Conclusions section was added.

Comment 6:  The NMR data was reexamined, and the designated error was corrected.

Round 2

Reviewer 1 Report

Although the authors took into consideration most of the comments, unfortunately the part about optimization (see my first comment) is still not fully disclosed. I recommend to expand the discussion part about optimization. 

Author Response

We thank the Reviewer 1 for continued attention given to experimental details.
Comments concerned with additional reaction details/optimization have been provided. Specifically,
crucial solvent effects for the metelloamination event have been given (manuscript lines 64-67).
Important solvent requirements have also been provided for the Cu(I) catalyzed alkynylation (lines 70-
71).
Should referee 1 have further specific inquiries in mind that need mentioning, these should be clearly
identified for us.

Reviewer 2 Report

The part of the manuscript related to results and discussions was improved and conclusions were added.

On the basis of the amendments and clarification of aspects of the work I am now pleased to recommend publication.

Author Response

We thank reviewer 2 for their recommendation.

Additional information on solvent effects has been added to the discussion in accordance with reviewer 1's request.

Reviewer 3 Report

The manuscript has been revised in accordance with the comments, and this referee happy to recommend it for publication.

Author Response

We thank reviewer 3 for their recommendation.

Additional information on solvent effects has been added to the discussion in accordance with reviewer 1's request.